# Automatically Generating Scenarios from a Text Corpus: A Case Study on Electric Vehicles

**Christopher W. H. Davis [1], Antonie J. Jetter [1] and Philippe J. Giabbanelli [2,*]**

[1]    Department of Engineering & Technology Management, Portland State University, Portland, OR 97201, USA;
      davis36@pdx.edu (C.W.H.D.); ajetter@pdx.edu (A.J.J.)
[2]    Department of Computer Science and Software Engineering, Miami University, Oxford, OH 45056, USA
[*]    Correspondence: giabbapj@miamioh.edu

**Abstract:** Creating 'what-if' scenarios to estimate possible futures is a key component of decision-making processes. However, this activity is labor intensive as it is primarily done manually by subject-matter experts who start by identifying relevant themes and their interconnections to build models, and then craft diverse and meaningful stories as scenarios to run on these models. Previous works have shown that text mining could automate the model-building aspect, for example, by using topic modeling to extract themes from a large corpus and employing variations of association rule mining to connect them in quantitative ways. In this paper, we propose to further automate the process of scenario generation by guiding pre-trained deep neural networks (i.e., BERT) through simulated conversations to extract a model from a corpus. Our case study on electric vehicles shows that our approach yields similar results to previous work while almost eliminating the need for manual involvement in model building, thus focusing human expertise on the final stage of crafting compelling scenarios. Specifically, by using the same corpus as a previous study on electric vehicles, we show that the model created here either performs similarly to the previous study when there is a consensus in the literature, or differs by highlighting important gaps on domains such as government deregulation.

**Keywords:** causal model; Fuzzy Cognitive Map; Q&A system; sustainability; text mining

## 1. Introduction

*What-if* questions are essential to making decisions by reasoning about the potential impacts of a situation. The situation could be an intervention (e.g., What happens to the sustainability of a city if we promote green spaces?) or a continuation of current trends (e.g., What happens in ten years if we continue with current emissions of pollutants?) [1]. A what-if question pertains to a specific *system*. For example, it would be impossible to answer the two questions above without a clear definition of the system (e.g., How do we measure sustainability? What is impacted by green spaces?). A *scenario* thus raises what-if questions of interest within the context of a clearly defined system, for example, by listing relevant factors and connecting them to track causal impacts. In other words, a scenario is a self-contained story about a potential future [2,3]. Scenarios have several demonstrated benefits for the decision-making activities of teams, such as raising awareness for the dynamics of an environment, managing uncertainty, evaluating different products, or breaking away from groupthink [4–7]. The field of *scenario planning* has articulated many approaches to craft such scenarios [8], often with the objective of producing a small number (typically 3–8) of plausible and alternative scenarios that cover different futures [9]. The quality of these scenarios is assessed through various criteria, such as plausibility [10], creativity [11], transparency [12], sufficient differentiation [13], relevance [14], or consistency [15].

A recurrent challenge is that scenario planning is a time-consuming and demanding process, for at least three reasons. First, the complexity of a system often calls for several

subject-matter experts (SMEs), who are identified and involved via a trained facilitator to shed light on driving forces and current trends. Comprehensively understanding a system can thus be a significant endeavor, mobilizing numerous SMEs and necessitating the availability of a trained facilitator [16,17]. Second, there may be disagreements among SMEs on how some aspects of a system operate, or such mechanisms may simply by unknown. Similarly, some existing trends in the system or the actions planned by other stakeholders may not be known. There is thus a need to represent uncertainty. Third, under many scenario-planning techniques, teams focus on the 'big picture' to assess the futures of entire markets, industries, or even societies. While this is useful for high-level strategical thinking, it does not address the needs of teams who need more granular information to make tactical decisions related to specific products.

Given these challenges, there has been particular interest in automating some or all of the process of scenario planning, resulting in *Foresight Support Systems* [18,19]. Text collections have been an essential data source for such systems [20], as an indirect way to obtain vast amounts of domain expertise. This reflects a broader trend in future studies, which leverages unstructured data from websites, news posts, or academic journals [21–24]. These text collections have primarily been analyzed through web scrapping and topic modeling; recent examples include [25–28]. However, none of these studies *fully* automated the end-to-end process of scenario generation. For instance, [26] manually map the system, and [27,28] manually perform desk research and verification. Even works leveraging advances in natural language processing (NLP), such as BERT, contain a manual step of risk identification [29]. In this paper, we posit that there is a potential to go further in leveraging the information connected through massive text collection by using NLP to extract models of the system and craft scenarios.

In this paper, we improve the automatization of scenario generation by combining natural language processing and Fuzzy Cognitive Maps (FCMs). Our proposed tool is named SAAM, for S̲cenario A̲cceleration through A̲utomated M̲odelling, and is available open source [30]. By emphasizing a fully automatic approach, we seek to drastically reduce the barriers to scenario development for teams who do not have the time or capacity to engage with subject-matter experts and trained facilitators.

To demonstrate the efficiency of our tool, we then apply it to a case study regarding electric vehicles (EVs). EVs were chosen as a guiding example for our technique as there is a demonstrated need and interest in scenario generation [31–33]. In particular, the scenarios covered by our case study include key themes about EVs, such as adoption [34–36], regulation and policy incentives [37–39], and technological enablers [40,41].

The remainder of this paper is structured as follows: To ensure that the manuscript is self-contained and usable both for computational scientists and sustainability specialists, our Background section provides the foundations for NLP and FCMs. Our Methods section builds on these foundations to introduce our proposed tool, SAAM. To demonstrate the efficiency of our tool, we then apply it to a case study regarding electric vehicles. Our results are compared with those obtained on the same corpus in a previous study performed by another group, showing that our model performs either similarly (with less manual involvement) or reveals important gaps. Our Discussion section contextualizes the potential of SAAM and outlines its limitations as well as opportunities for future improvements.

## 2. Background

### 2.1. Fuzzy Cognitive Maps

As evoked in the introduction, a scenario exists within the context of a clearly defined system. In other words, we need to *model* this system. Suitable modeling approaches fall into two broad categories. *Conceptual models* (e.g., causal maps, causal loop diagrams, mind maps) provide a structure to the system by identifying relevant factors and their interconnections [42–44]. Conceptual models have several benefits, such as identifying key factors in a system (e.g., via centrality), revealing themes (e.g., via community detection), or comparing perspectives (e.g., via Graph Edit Distance) [44–46]. However, these

models offer limited support for scenario planning. For example, we can ask *what* will be impacted in a scenario, and we will follow links in the model to provide a list (e.g., via a Breadth-First Search). However, there is no quantification; hence, we cannot say whether some elements will be impacted more or less. In other words, the inability of a conceptual model to provide a quantitative estimate limits the decision-support tasks for which they are suitable. The second category of *quantitative (aggregate) models* offers these capabilities, but building them requires significantly more work [47]. Quantitative models are *simulation* models, which means that they can provide numerical answers by updating values based on certain rules. A well-known quantitative approach is System Dynamics [48], where the model runs differential equations to update concepts based on rates over time; this approach can provide highly accurate point-estimates, but requires significant quantitative data. Fuzzy Cognitive Maps (FCMs) do not include the notion of time; hence, they are simpler to build (e.g., entirely from qualitative data) at the expense of lower accuracy (i.e., cannot know exactly *when* an effect will be obtained) [49]. FCMs have been used in over 20,000 studies [50], including many works on scenario planning, as they provide quantitative system models that suffice to represent the driving forces that shape the future (e.g., technology, economy, social trends) and their interdependencies. Recent examples in sustainability include modeling the wind energy sector [51,52], social sustainability [53,54], planning viewed by rural communities [55] or urbanites [56], or managing waste flows [57]. Throughout these examples, the FCM is used for simulations by varying the input values to produce multiple scenarios; since the scenarios are all based on the same model, they are guaranteed to be internally consistent.

Mathematically, an FCM has two parts: a *causal structure* (similar to a conceptual model) and an *inference engine* (to run simulations). The causal structure is represented as a directed, weighted, labeled graph $G = (V, E)$, where $V$ is the set of labeled nodes and $E$ is the set of directed edges. Both nodes and edges have a weight. The weight of each node changes over each simulation step $t$ to denote the extent to which a concept is present (1) or absent (1); it is denoted by $v_i^t \in [0, 1]$. The weight of each edge is held constant as it is considered a property of the system (e.g., if there are many anglers, then there are much less fish), whereas nodes correspond to a case (How many fish are there at a given point?). Edges are represented with an adjacency matrix, where $W_{i,j} \in [-1, 1]$ indicates the weight from node $i$ to $j$. The weight is 0 if there is no relationship, positive if an increase in $i$ causes an increase in $j$, and negative if an increase in $i$ causes a decrease in $j$. The inference engine operates by synchronously updating all the nodes' values per Equation (1):

$$v_i^{t+1} = f\left( v_i^t + \sum_{j \in V, j \neq i} W_{j,i} \times v_j^t \right) \tag{1}$$

Intuitively, this update means that the next value of a node accounts for its current value (i.e., there is memory for one step), as well as the values of all incident nodes and the corresponding causal strengths. The function $f$ serves to keep the output in the desired range $[0, 1]$. The update is performed until a stopping condition is met. The *desired* stopping condition is that a set of key nodes $O$ (considered as outputs of the system) change by less than a user-defined value $\varepsilon$ between two consecutive iterations. It is possible that this desired situation is not reached, due to oscillations or chaotic attractors. To ensure that the algorithm stops in any case, a secondary condition is a hard cap on the maximum number of iterations $\tau$. Consequently, the updates stop if and only if Equation (2) holds true [58]:

$$\forall o \in O, \left| v_o^t - v_o^{t-1} \right| \leq \varepsilon \ or \ t \geq \tau \tag{2}$$

As the mathematics of FCMs have been abundantly covered elsewhere, we refer the reader to seminal reviews for further details [59,60]. In this paper, our interest is on (i) generating FCMs from text, and (ii) using them to craft scenarios. With regard to (i), we note that several works have extracted causal maps from text [26,61–63]; hence, they could

generate the causal structure, but did not produce a complete FCM. Some works have focused on creating FCMs from summaries or large collection of documents [64,65], but they needed manual interventions (e.g., manual labeling, expert verification); hence, the process was only semi-automatic. The objective of (ii) building scenarios with FCMs is pursued by many studies [66–68], with several examining the role of FCMs as a communication tool to engage stakeholders in scenario generation [69,70].

### 2.2. Natural Language Processing

The major companies that own big data (e.g., Microsoft, Google, Amazon) have heavily invested in model creation and made several of the resulting models available to researchers and practitioners through their web services. For example, Google provides pre-trained models for natural language processing via its Natural Language AI. Pre-trained models in NLP often leverage deep neural networks, resulting in highly used models such as BERT or GPT [71,72]. BERT is of particular interest here, as it has previously been used to extract causal models from text [29]. We recently described BERT as follows [73]:

"BERT is a pre-trained deep bidirectional transformer, whose architecture consists of multiple encoders, each composed of two types of layers (multi-head self-attention layers, feed forward layers). To appreciate the number of parameters, consider that the text first goes through an embedding process (two to three dozen million parameters depending on the model), followed by transformers (each of which adds 7 or 12.5 million parameters depending on the model), ending with a pooling layer (0.5 or 1 million more parameters depending on the model). All of these parameters are trainable."

Intuitively, BERT models are trained by first creating a base model on a large unstructured dataset that can make predictions such as what word might appear next in a sentence. Secondly, the previous learnings are transferred, and models are fine-tuned on specific datasets that allow such functionality as answering questions based on the text in the dataset. To achieve this, BERT uses multiple layers of encoding so it can predict context and "understand" the difference between semantically similar terms such as "apple pie" or "apple tree" by encoding (1) the words, (2) the sentences, and (3) the positions of the words in the text. This combination of tokens is then fed into a neural network that creates the base model, which can be fine-tuned on specific text for NLP tasks. For a more detailed description of BERT, we refer the reader its highly cited source [74].

The core idea of repurposing BERT to extract a causal model is to build a question-answering (Q&A) system [75] in which we ask the question of what 'causes' or 'results' from a given factor, and then repeat the process on these causes and consequences to gradually build a model. In other words, a Q&A system can determine connections and causality between concepts in the model. By asking the system, "why do people buy more electric cars?" a human user identifies a concept of interest through the question—in this case, "electric cars". Q&A systems provide the answer by treating a pre-selected text corpus as the context. In this example, the corpus would focus on the electric car industry.

To briefly illustrate this notion within the context of sustainability, consider the following example of the fashion supply chain and the guiding question, "What causes pollution?" By applying a Q&A BERT-based model from the Hugging Face project [76] on online books about the fashion supply chain, we obtain a sample output such as in Table 1. Items in the 'answer' columns are concepts, the 'confidence' is the degree of certainty with which the algorithm identified the answer, and the 'context' provides an excerpt from the most relevant document containing the answer. In this example, "fast fashion brands" is returned with high confidence because it is directly referenced in the text as a cause of pollution, whereas very low confidence was returned for the other concepts because they are mentioned together but do not answer the question based on the text provided. The more text that associates fast fashion brands with pollution, the higher the confidence value would be. The context can also help to identify more relevant concepts, which can be used for further questions [77]. For instance, 'sustainable development' is mentioned as part of the answer 'global climate change', and it could lead to another line of questioning by

asking the Q&A system, "What types of sustainable development are happening in the fashion industry?".

**Table 1.** Sample output from an NLP Q&A system when asked, "What causes pollution to increase?".

| Answer | Confidence | Context |
|---|---|---|
| Fast fashion brands | 0.489 | on the other hand, fast fashion brands such as h & m, Zara, Topshop, have been blamed for creating poor labor welfare, severe environmental pollution as well as a massive amount of clothing disposal at the end of the product life cycle. |
| Global climate change | 0.00713 | introduction due to the aggravation of environmental pollution and global climate change, sustainable development has attracted more and more attention. |
| Overconsumption of energy | 0.00669 | by doing so, these companies alleviate conflicts of interest among participants and reduce pollution and overconsumption of energy. |

## 3. Design of the Proposed SAAM tool

*Overview*

Our work seeks to automate the process of scenario generation. However, the analysts still need to be involved in defining the question and pointing to acceptable data sources. From that point onward, the automatic process can run. Overall, our proposed SAAM tool is composed of three stages: setup (which is manual), model building (which is automatized), and model use by humans (Figure 1); each of these stages is explained in a dedicated subsection below. Several parameters are involved in these stages, as summarized in Table 2. In short, the automation collects the data, runs the Q&A algorithms to find traceable answers from the text corpus, and builds the initial model as a Fuzzy Cognitive Map. People can inspect the answers, define filters, and potentially ask more questions to build out the model further. Once the model is fully built, people use it to run their scenarios. This process promotes an interplay of human interaction and Artificial Intelligence, hence following the human-in-the-loop approach that is increasingly promoted in machine learning to create more explainable models [78,79].

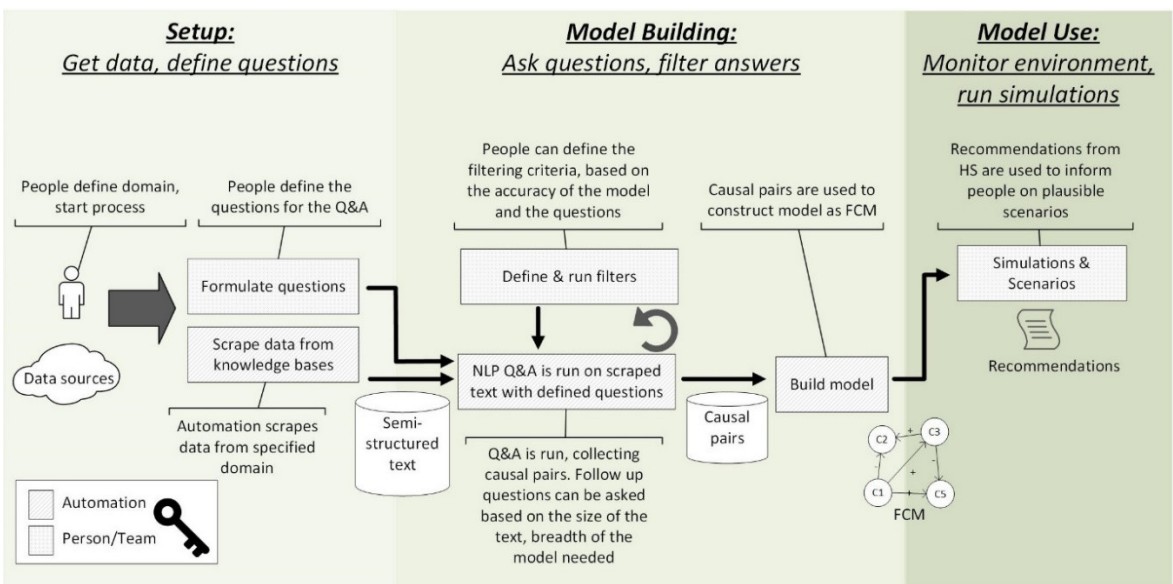

**Figure 1.** Overview of our proposed SAAM system.

**Phase 1: Setup by defining questions and identifying relevant data sources.**

Depending on the application domain, the modeling team starts by determining the questions to ask. This does not depend on their computational knowledge. It may depend

on the stakeholders and commissioners, as is the case for any modeling endeavor [80]. For instance, if the modeling team seeks to better understand the future of self-driving vehicles, then they may ask questions that contain key terms such as "self-driving", "vehicles", or "self-driving cars". That is, they are responsible for identifying a set of seed concepts (or "nodes" of an FCM) belonging to the domain. If the modeling team is unsure about keywords that characterize a domain, they can also use NLP on relevant documents to extract candidate keywords, for instance, by removing stop-words and then extracting keywords with high frequency using libraries such as RAKE or Gensim. The keywords need to be structured into a question that can be passed onto a Q&A system. Two main options are as follows: If the team seeks a model to perform cause-and-effect analyses, then they may start with questions such as "what causes [phenomenon] to increase" and its complementary "what causes [phenomenon] to decrease"; this is similar to a facilitated modeling process investigating risks and protective factors [43]. Alternatively, if the team seeks a model that explores drivers for a specific technology, then they define questions based on the Political, Economic, Social, Technical, Environmental, and Legal (PESTEL) aspects of the technology. The PESTEL framework has been commonly used in scenario planning [81,82] and will be exemplified in our case study.

The modeling team also identifies appropriate data sources. These may include journal articles, newspaper articles, or websites that provide detailed information for the target domain.

**Phase 2: Model building through the Q&A System and filtering.**

The modeling team is responsible for specifying the number of iterations through which the system should build a model (i.e., 'question depth'). For example, after finding that A causes B, the model could be expanded to know what causes B, leading to another round of questions on increasing and decreasing causes of B; this would constitute a question depth of 1 (Figure 2). A modeler may choose a higher question depth if they only have a single question to start with, or if the corpus used is very large. After a certain number of iterations, answers typically start to decrease in confidence because they reach the knowledge limits of the corpus.

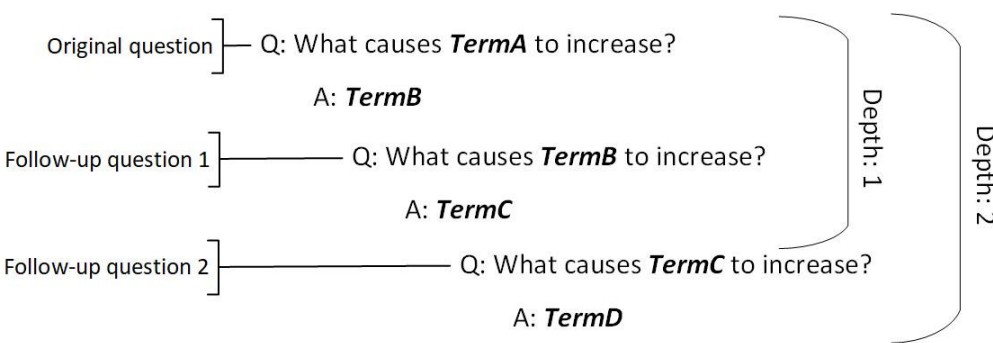

**Figure 2.** The 'question depth' is a key parameter governing model complexity.

Given (i) the corpus and (ii) the set of questions originating from phase 1, as well as (iii) the question depth, we use an NLP Q&A system to repeatedly find connections between concepts. Our work specifically uses the Hugging Face Q&A pipelines, but implementations can also be achieved via other open-source solutions such as Sentence Transformers [83]. When a factor X is identified as increasing Y, then we create an edge from X to Y with the value 1; conversely, if X decreases Y, then the edge has the value −1. Tracking the polarity of the relationship is important to later create the FCM.

Similar to the example in Table 1, the Q&A system responds to each question by providing the answer, together with a confidence level between 0 and 1, indicating the probability that the model got the correct answer, and token markers indicating where in the document the answer was found. For example, if a document contains the sentence "Pollution is a direct cause of a lower standard of living," and the Q&A algorithm asks the

question, "what causes lower standards of living?", the model will return "pollution" as the answer, a high probability such as 0.89, and the beginning position in the document to where the answer was found. From these values, the answer and confidence score are directly relevant to assisting the modeler, and the token marker can be used to find the sentence and the document the answer was found in to give people using SAAM the full context of the answer. In this example, it is as if the model is saying "I am pretty sure that pollution is the answer because of this excerpt from the text you showed me". If responses were unfiltered, three problems could occur. First, answers with low confidence could be included, resulting in *noise* in the model. Second, words that look different but actually have the same meaning would be kept separately, hence resulting in a seemingly comprehensive but actually *redundant* model. Third, the name of a concept is usually a noun, but answers may consist of other types of words such as adjectives, which would be *harder to interpret* as labels in a causal model (e.g., the noun 'height' would be preferable to the adjective 'tall').

We handle these three situations through three filters, whose values can be set by the user. First, to avoid noise, the modeler may only keep connections that were returned with a high degree of confidence, thus filtering out results whose confidence is below a user-defined *confidence threshold*. The threshold depends on the Q&A model used and the corpus; hence, it should only be determined by the modeler after reviewing the initial results. Second, to avoid redundancy, the user provides a *semantic similarity threshold* between concepts such that answers above this value are deemed similar and merged. The semantic distance can be defined using Levenshtein or cosine distances. Our implementation uses the Levenshtein distance provided by the fuzzywuzzy library in Python [84], where a threshold of 100 is an exact match, and the closer to 0, the larger the distance between words. Finally, Part Of Speech (POS) tagging gives us the type for each word, and the user can *filter out POS* that do not belong to a causal model. We use the spaCy library [85] for this purpose. The default filter removes adjectives, punctuation, particles, symbols, and interjections.

**Table 2.** Parameters and inputs to our proposed SAAM system. Additional libraries were used for the system as a whole: Azure Machine Learning for data hosting and computing, and Machine Learning Pipelines for coordination of tasks.

| Parameter/Input | Values | Purpose | Libraries Involved |
|---|---|---|---|
| Seed questions | String | The modeling team must define the problem of interest, which anchors the model. | N/A |
| Text collection | Resource set | Natural language processing is performed over a text corpus. It can be provided directly (e.g., as files or URLs) or retrieved from databases with search keywords. | Power Automate Desktop (to automate data collection) |
| Confidence threshold | [0, 1] | Filter results based on the confidence returned by the Q&A algorithm. The threshold range will vary based on the context; thus, the cut-off is up to the modeler. | Hugging Face Q&A |

**Table 2.** *Cont.*

| Parameter/Input | Values | Purpose | Libraries Involved |
|---|---|---|---|
| Semantic similarity threshold | [0, 100] (100 indicates perfect match) | Combine concepts that are semantically similar. This can use Levenshtein or cosine distance. A lower value will group many concepts, a higher value may create a model with different concepts but similar meanings. | fuzzywuzzy |
| POS Filtering | Array of POS tags (universal POS tags [86]) | Parts of speech may be returned as answers, but would not make intuitive sense as concepts. In addition, aggregate models often limited to only using nouns as concepts. | spaCy |

**Phase 3: Using the model.**

Phase 2 produces a model in the form of a Fuzzy Cognitive Map. As explained in our background, scenarios can be built using this FCM, based on situations that are currently considered by stakeholders. This is illustrated in the next section through our application of SAAM to electric vehicles.

## 4. Methods: Applying SAAM to Study Electric Vehicles

### 4.1. Overview

Our case study demonstrates the ability of our proposed SAAM system to extract concepts and causal links from a text, structure them into an FCM model, and use the model to run simulations on alternative future scenarios that are plausible, decision-relevant, and cover the range of uncertainty. For a fair comparison of the results obtained by SAAM, our case study follows the published work of another research team, such that we have matching objectives (study of electric vehicles), but different techniques. Specifically, the prior work used the PESTEL framework, followed by Latent Semantic Analysis (LSA) and Fuzzy Association Rule Mining to build a model *semi*-automatically [87]. The differences between their work and our approach are visually summarized in Figure 3. Most importantly, concept mapping was a manual endeavor in the previous study, while our work seeks to automatize this task as part of model building. Consequently, our comparison of SAAM's output with the previous study seeks to determine whether a more automatic approach can yield a similar model. Our workflow is summarized in Figure 4 and detailed in the following subsections.

### 4.2. System Setup: Data Sources, Seed Questions, Parameters

The authors of the comparison study did not publish the data they used. Consequently, we reconstructed the datasets from their description. Specifically, they scraped five websites: Siemens [88], MIT Technology Review [89], Kurzweil Accelerating Intelligence [90], World Future Society [91], and FutureTimeLine [92]. These sites were used by the authors of the prior work because they all provided articles that were future-oriented, hence, already containing an analysis of trends and expert insight on potential futures. Note that the prior work was published in 2016; hence, it would not be a fair comparison if we built a model based on the data available up to today (2022). In addition, some of the websites have ceased to exist, hence content may not only have expanded but also have been deleted. Consequently, we used the web archive Wayback Machine to re-create a dataset that most closely resembles the content available to authors of the prior work [91]. Specifically, we (i) only scraped articles discussing electric vehicles or alternative energy, as this filter was

noted by the authors of the prior work; and (ii) we used the Wayback Machine to scrape data that would have been available as of March 2016.

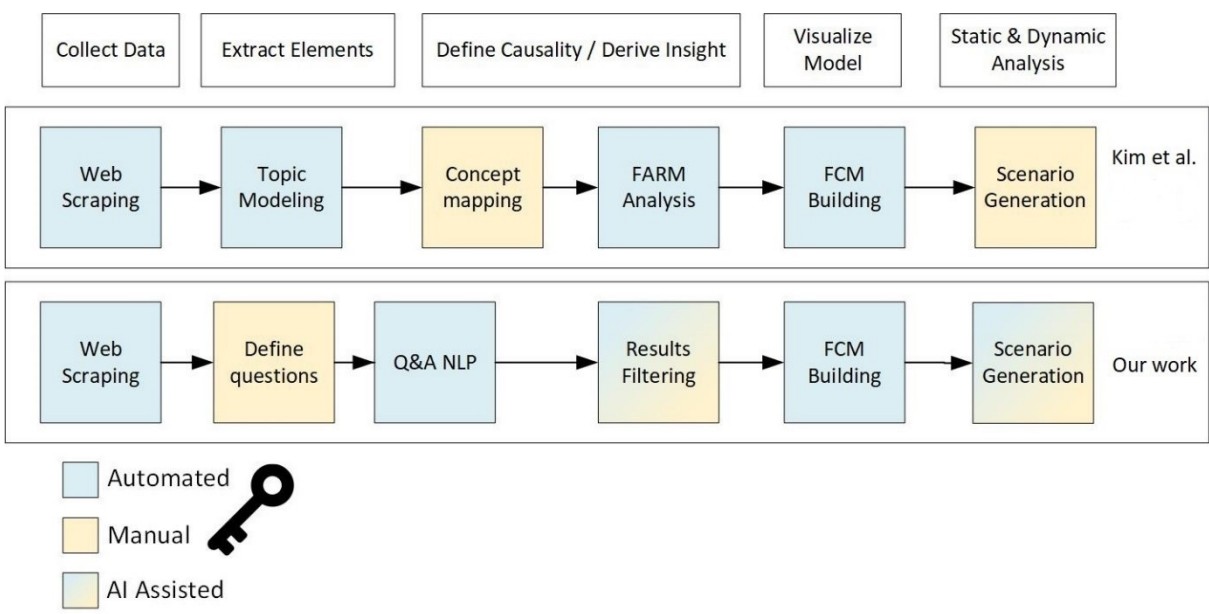

**Figure 3.** Comparison between our study (bottom row) and the prior work of Kim and colleagues [87].

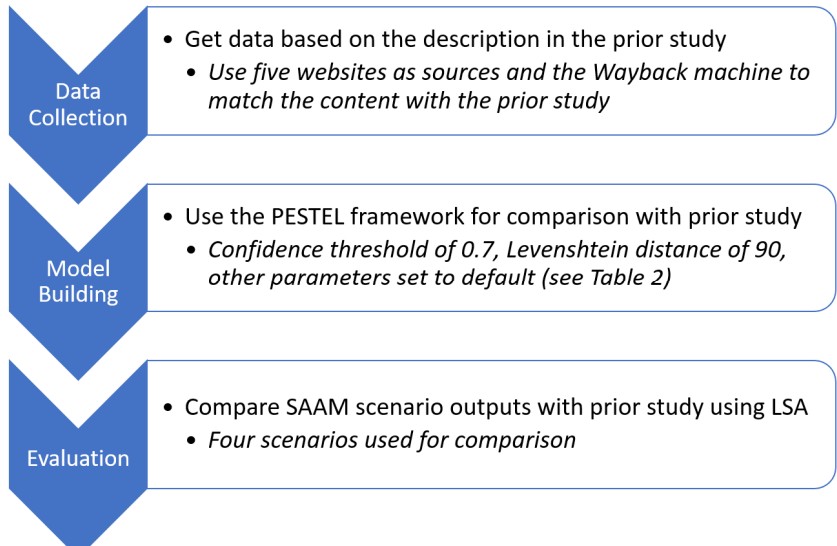

**Figure 4.** Key steps and specifications for our comparative study on electric vehicles.

Since the prior work used the PESTEL framework for its guiding questions, we also started by creating a set of questions about electric vehicles for each aspect of PESTEL. For example, under the environmental category, we asked, "What are benefits to the environment" and its complementary "What hurts the environment?" The full list of seed questions for our Q&A system is provided in Table 3. Parameter values for SAAM are listed in Figure 4.

**Table 3.** Seed questions based on PESTEL for our comparative study on electric vehicles.

| Question | Subject | Weight | PESTEL Category |
|---|---|---|---|
| What technology is needed for electric vehicles? | EV adoption | 1 | Technology |
| Why use an electrified vehicle? | | 1 | Open |
| What are impediments? | | −1 | |
| What are political factors? | | 1 | Political |
| What are the benefits to the environment? | | 1 | Environmental |
| What hurts the environment? | Environment | −1 | |
| What are social benefits? | EV adoption | 1 | Social |
| What are social problems? | | −1 | |
| What social aspects affect electric vehicles? | | 1 | |
| What are the economic benefits? | Economy | 1 | Economic |
| What are economic problems? | | −1 | |
| What are economic drivers? | | −1 | |
| What are legal problems? | EV adoption | −1 | Legal |
| What are legal drivers? | | 1 | |
| What are legal benefits? | | 1 | |

*4.3. Comparison: Model Content and Simulated Scenarios*

Models can be compared on the basis of their structure (e.g., which variables do they include? How are they connected?) and outputs (e.g., given the same input, which results do they produce?).

To compare the structure of the models, we examined the terms that they contained. To guide the comparison, we grouped the content of the SAAM model using the same categories as in the prior work. We stress that *our objective is not to find models with the same structure*. Rather, the structural comparison can tell us whether the models include similar categories, or aspects where one model was more comprehensive than the other. In contrast, we do *expect more similarities when comparing the output* of the models. For each scenario, we ran the SAAM model by creating inputs corresponding to the ones used in the original study, and then we compared the outputs of the two models. The original study had four high-level scenarios: (1) application of EV to tourism, (2) failure to develop battery technology, (3) failure of EV adoption in general, and (4) relaxation of government regulation. Changes were necessary in our comparative study, for two reasons. First, the prior work grouped the terms "economy", "consumer", "customer", "growth", and "tourism" in the tourism category by assuming that tourism is driven by consumers and is directly related to the economy. To avoid this narrow assumption, we broadened the scenario to study economic factors. Second, scenarios (1) and (3) are actually linked because (1) studies the effects of widespread EV adoption, whereas (3) examines the failure of widespread EV adoption. If we performed two scenarios on the same aspect, then that specific aspect of the model would artificially be counted twice. Consequently, we ran simulations on three scenarios: (i) economic factors affecting EV adoption, whether the economy is good or bad; (ii) what happens if battery technology does not develop; and (iii) what happens if the government decides to not help the EV industry at all by removing any incentives for EV and stopping any regulation efforts to increase adoption.

## 5. Results

### 5.1. Structural Comparison: Content of the Models

After filtering, the model produced by SAAM resulted in 52 unique concepts with 110 connections, as compared to the 15 concepts and 44 connections from the original study. The terms identified are shown in Box 1. As described in the previous section, we start our comparison by applying the categories from prior work to group the terms found by SAAM. The comparison is shown in Table 4. SAAM identified some of the same terms that were identified in the original study (green highlights), but also found concepts that were not detected in the prior work. For example, SAAM identified aspects such as consumer confidence, infrastructure investments needed, and natural resources required to build required batteries. This more comprehensive assessment can provide deeper insight into the data and hence support the creation of more robust models. Asking specific questions about social impacts led to answers such as 'thinking globally and acting locally', which was not in the LSA method. On the other hand, a few of the topics identified only make sense when knowing the context; for instance, 'your gas guzzler' refers to today's cars that run on gas, while 'aboriginal training' came from an Australian article about retraining individuals from underserved communities to work in new jobs created by the electric vehicles industry. Note neither the list of terms identified by SAAM nor those covered in the original study claim to address every facet of electric vehicles; rather, they extract information from a corpus focused on technology development. For example, emerging aspects such as electric mobility education [93] were absent from the corpus; hence, they are also absent in the list of terms.

### 5.2. Scenario Comparisons

Numerical results for each scenario are provided in the Table A1. Note that in the deregulation scenario, results are only indicative since the system oscillates instead of reaching stable values.

The original study showed that applying EV to tourism resulted in increased employment, a better economy, lower pollution levels, and improved energy efficiency. However, none of the data had articles about tourism; hence, the SAAM model did not directly cover tourism. After noting that the original study grouped tourism with economic benefits (see Section 4), we broadened the **scenario to the economy**. Specifically, we set the constructs 'employment', 'business development', 'current unit sales', 'economic activity', 'economic and safety benefits', and 'wealth' to high in one case (good economy) and to low in the other (poor economy). The SAAM model output a different result than the original study, noting that in a *good economy* 'no exhaust emissions' are adopted, but 'greenhouse gas emissions' increase and negatively affect 'the air'. In addition, we got richer results with SAAM, through some of the concepts that were not identified in the prior work; for instance, 'think globally act locally' decreases in a good economy, 'public investment' increases, and 'lack of infrastructure' decreases (meaning that the infrastructure will start to improve). In a good economy, 'EV adoption' decreases and 'your gas guzzler' (representing existing gas-powered vehicles) increases. In a *bad economy*, the inverse happens. Although this may seem counterintuitive at first, the transparency of the SAAM model lets us realize that, while several variables (technology, consumer confidence, battery technology) are high, the focus on sustainability decreases and volatility in gas prices decreases, which ultimately hurts the adoption of EVs. In short, this scenario implies that in a good economy, several technological aspects improve (EV infrastructure, battery technology, energy efficiency), but there is no strong *drive for consumers to adopt* EV technology.

**Table 4.** Comparison of concepts found by our SAAM system with the prior work's use of LSA. Categories are taken from the prior work to facilitate the alignment of the two models. Simple matches are shown in green, while noting that additional terms are equivalent within this context.

| Category | SAAM Concepts | LSA Concepts |
|---|---|---|
| Air pollution | greenhouse gas emissions, no exhaust emissions, the air, your gas guzzler, energy pollution | Temperature, environment, pollution, atmosphere, carbon dioxide emission, greenhouse gas, $CO_2$, eco |
| Alternative energy technology | clean renewable energy sources, polarization systems | Renewable energy, diesel, biofuel, biomass, geothermal, petroleum, gasoline, hybrid, photovoltaic, solar energy |
| Battery technology | batteries, power and mileage limits, recharge speed | Lithium battery, ion battery, acid battery, storage, battery life, lightweight, BMS, lithium ion battery |
| Charging technology | a comprehensive charge station network, generic supercharging stations | Wireless power, charger, recharge, power transmission, charger |
| Costs reduction | EVs cost, the falling price of batteries, incentives | Cost reduction, incentive, support, maintenance cost |
| Economic revenue | business development, current unit sales, wealth, economic activity | Economy, growth, sales, investment, revenue, GDP, trade, import, export |
| Energy efficiency | energy efficiency | Energy efficiency, energy consumption, efficiency improvement, energy density, mileage |
| Government regulation | carbon pricing, cities conservation, governments, incentives, public investment, regulation | Regulation, incentive, policy, government, limitation, standard, tax reduction, policy |
| Industry-university collaboration | scholarships, aboriginal training | Company, startup, university, laboratory, investment, partnership, entrepreneur, grid |
| Job creation | employment | Job, worker, manufacturing, services, employment |
| Motor technology | electric motor | Engine, inverter, magnet, DC, AC, torque, capacity, motor |
| Usability | information technology | Automation, sensor, network connection, software, comfort, assistant, internet |
| Public transportation | Self-driving vehicles | Transportation, electric bus, driver, passenger |
| Safety | economic and safety benefits, self-driving vehicles | Safety, driverless, collision, vibration, pressure, security, stability, obstacle warning, monitoring |
| Other | thinking globally and acting locally, a completely carbon neutral transportation option, biomimicry, confidence, durability, environmentally conscious citizens | |
| Application to tourism | | Consumer, customer, tourism, growth, economy |

**Box 1.** List of terms identified by SAAM.

'EV', 'a completely carbon neutral transportation option', 'a comprehensive charge station network', 'aboriginal training', 'artificial intelligence', 'batteries', 'biomimicry', 'business development', 'carbon pricing', 'cities conservation', 'clean renewable energy sources', 'confidence', 'current unit sales', 'durability', 'economic activity', 'economic and safety benefits', 'electric motor', 'employment', 'energy efficiency', 'energy pollution', 'environmentally conscious citizens', 'evs cost', 'fear', 'gaps', 'generic super charging stations', 'governments', 'greenhouse gas emissions', 'harmony', 'incentives', 'information technology', 'infrastructure', 'lack of hydrogen infrastructure', 'liability', 'no exhaust emissions', 'oil and gas volatility', 'polarisation systems', 'potential roadblocks', 'power and mileage limits', 'public investment', 'rare earth metals', 'recharge speed', 'regulation', 'remote communities', 'save lives', 'scholarships', 'self-driving vehicles', 'significant technology improvements', 'sustainability', 'the air', 'the falling price of batteries', 'the power and mileage limits', 'thinking globally and acting locally', 'traffic congestion', 'transform mobility', 'wealth', 'your gas guzzler'

In the **scenario where battery technology fails to develop**, the original study concluded that there will be less job creation, less tourism, a poor economy, and an increase in pollution. To investigate this scenario, we set the corresponding variables in our model to low: 'batteries', 'lithium-air batteries', 'lithium-ion', 'lithium-ion batteries', 'recharge speed', 'power and mileage limits', and 'energy efficiency'. SAAM also found that 'employment' decreased, and terms associated with the economy ('economic activity', 'business development', 'current unit sales', 'wealth') all ended on low values. However, as in the previous scenario, SAAM had an inverse relationship between the economy and the environment; hence, it forecasted a decrease in 'greenhouse gas emissions' with an accompanying increase in the quality of 'the air'. In this scenario, EV adoption starts to improve even though the cost of EVs ('EVs cost') is driven up. Although battery technology fails to improve, an increased desire for sustainable solutions ('sustainability') and growing investment from the government ('public investment') help to offset the high cost of EVs.

Finally, in the **scenario of relaxing government regulations**, the prior work concluded a reduction in costs, an increase in safety, and an increase in energy efficiency. We simulated this scenario by setting all relevant concepts to low ('regulation', 'incentive', 'policy', 'government', 'limitation', 'standard', 'tax reduction'). Our simulation produced a limit cycle rather than a stable state. This indicates that if the government does nothing, then consumers would oscillate between EV adoption and rejection as the environment shifts from one preference to another based on competing factors. This sensitivity of our model to regulation suggests that it is a key concept in the adoption of EVs; hence, it deserves particular consideration when examining future strategies.

## 6. Discussion

### 6.1. Findings and Implications

Examining future scenarios is necessary to support decision-making activities [4–7]. These scenarios are created by teams and run on quantitative causal models, which forecast potential effects based on the evidence base. Creating a model is thus the cornerstone of scenario generation, yet it has long been a labor-intensive task [8,9]. Several works have brought automation to this process [18,19], in particular by deriving models from an evidence base consisting of a text corpus [25–28]. The recent work of Feblowitz and colleagues at IBM [29] is the closest to our approach in numerous regards: starting from a set seed of concepts (or 'risk forces'), it automatically fetches documents (multiple times daily via the Watson Discovery service) and uses a Q&A system powered by Hugging Face's Transformers to extract a model, noting when concepts can be deemed equivalent. A key limitation in previous works is that several steps continue to be performed by humans, as is the case in [29] where (meta)data on causal relationships is obtained via a crowd-sourced questionnaire, whereas we use the weights from the Fuzzy Cognitive Maps. In this paper, we proposed a step further in automation by only asking the modeling team to provide the initial guiding questions and the evidence base, and then creating a model. We demonstrated that the model could be used to investigate scenarios, by focusing on a

case study in electric vehicles (EVs). EVs were chosen as a guiding example since (i) they have been the subject of several studies involving carefully crafted scenarios [31,94], and (ii) a previous study [87] with partial automation offered a direct comparison point with the model produced by our approach.

There are two key differences between our proposed approach (SAAM) and the prior study, which used less automation and involved Latent Semantic Analysis (LSA). First, LSA is used to find topics in a text collection and group terms together. Our system is not designed to perform such grouping, as we instead focus on finding terms by asking direct questions. The models are thus structured differently, with more granular content in SAAM offering a larger number of factors. However, it is possible that some of the content becomes too granular and needs to be interpreted given the context (e.g., 'the air'). Second, our proposed method and the previous one both have parameters that should be tuned by users. However, the methods are different; hence, the parameters offer control on different aspects. In SAAM, the modeling team can control filters, for instance, to force a simplification of the model by (i) combining semantically similar concepts and/or (ii) only accepting concepts where the system has high certainty. In contrast, the LSA method requires people to set a topic cluster size and manually name each final topic. Although our machine learning algorithm requires some human intervention to set parameters, we note that involving humans to train algorithms has been shown to facilitate co-learning between people and computers [95], and give analysts a better overall understanding of the model [96]. The potential benefits of a human-in-the-loop approach are noteworthy since our work is based on BERT, which is part of the set of artificial neural networks that have historically been characterized as 'lacking interpretability' and hence faced drawbacks in terms of trustworthiness by human decision makers [97].

Scenarios are supposed to help us step back and see the bigger picture, think outside the box, and consider alternatives that might not be obvious. Our results have shown that SAAM was able to generate alternative future scenarios that met this objective. We also demonstrated that the scenarios created via SAAM often agree with those created in the prior study, or propose a plausible line of reasoning when results differ. We emphasize that the application to electric vehicles provided a thorough evaluation of SAAM, but our tool is not limited to this specific application as it constitutes a reusable approach to generate scenarios. SAAM could thus be applied to similar issues in sustainability, such as autonomous vehicles [98], which have already been the subject of scenario generation studies using Fuzzy Cognitive Maps [99]. Our tool can more broadly benefit areas that frequently engage in the development of data-informed scenarios [100,101].

### 6.2. Limitations and Opportunities for Future Studies

One limitation of our comparison was the inability to use the same data as the original study, since it did not publish it. We re-created a dataset based on the sources and selection criteria mentioned, and ensured that it reflected what was available to the authors at the time. However, we did not detect any application to tourism in the evidence base; hence, this aspect was missing from the model and ultimately the scenario based on tourism was broadened to the economy.

The inspiring work by Feblowitz and colleagues suggests several improvements [29]. In particular, they were able to automatically generate trajectories from their model, using a planner and a clustering algorithm. To the best of our knowledge, planners able to generate a set of high-quality solutions (i.e., top-k planners) have not been applied to Fuzzy Cognitive Maps; hence, such algorithms would have to first be developed before we can produce trajectories.

The ability to transparently examine how the model reached a certain conclusion also holds particular promise for future studies. Indeed, the socio-environmental systems examined in sustainability studies are often complex, and models are at risk of becoming a 'black box' by being almost as complex. Maeda and colleagues stressed that "as the increasing complexity of models starts to influence policy making, it is important for scien-

tists to create new approaches to communicate their underlying assumptions, reasoning, data and methods to stakeholders" [102]. Future work can thus contribute further to this communication component, for instance, by leveraging the Q&A system not only to build the model but also to ask how conclusions were reached.

## 7. Conclusions

Generating scenarios is essential for decision-making activities, but it involves a labor-intensive step of model building. We proposed a system (SAAM) that goes beyond previous automation initiatives, and we demonstrated that the system can result in well-formed scenarios by contrast to a previous study on electric vehicles. As the first manuscript detailing and applying SAAM, there are several opportunities for future work in improving components of the system or applying it for other fields of sustainability that heavily depend on scenario generations.

**Author Contributions:** C.W.H.D. developed the software and wrote the first draft. P.J.G. edited the manuscript. A.J.J. and P.J.G. supervised C.W.H.D. All authors approved the manuscript. P.J.G. revised the manuscript. All authors have read and agreed to the published version of the manuscript.

**Funding:** This research did not receive any specific grant from funding agencies in the public, commercial, or not-for-profit sectors.

**Institutional Review Board Statement:** Not applicable.

**Data Availability Statement:** The code is available on our repository [30].

**Acknowledgments:** We thank Charles Weber and Ameeta Agrawal at Portland State University for their helpful feedback.

**Conflicts of Interest:** The authors declare no conflict of interest.

## Appendix A

**Table A1.** Each scenario is designed by setting the values of relevant factors in the model. For each scenario, we note the effect on other variables, as well as on the key construct of adopting electric vehicles (bottom row).

| Categories | Concepts | Bad Economy | Good Economy | Battery Fail | Deregulation |
|---|---|---|---|---|---|
| air pollution | greenhouse gas emissions | −0.952398323 | 0.952398323 | −0.957583063 | −0.959324401 |
| air pollution | no exhaust emissions | −0.952398323 | 0.952398323 | −0.957583063 | −0.959324401 |
| air pollution | the air | 0.957140415 | −0.957140415 | 0.937957076 | 0.957801403 |
| air pollution | your gas guzzler | −0.952398323 | 0.952398323 | −0.957583063 | −0.959324401 |
| alternative energy technology | clean renewable energy sources | 0.156727117 | −0.156727117 | 0 | 0.121182442 |
| alternative energy technology | polarisation systems | −0.952398323 | 0.952398323 | −0.957583063 | −0.959324401 |
| battery technology | batteries | −0.691699732 | 0.691699732 | −1 | −0.646327877 |
| battery technology | power and mileage limits | 0.156727117 | −0.156727117 | −1 | 0.121182442 |
| battery technology | recharge speed | 0.156727117 | −0.156727117 | −1 | 0.121182442 |

**Table A1.** *Cont.*

| Categories | Concepts | Bad Economy | Good Economy | Battery Fail | Deregulation |
|---|---|---|---|---|---|
| other | thinking globally and acting locally | 0.156727117 | −0.156727117 | 0 | 0.121182442 |
| charging technology | a comprehensive charge station network | −0.952398323 | 0.952398323 | −0.957583063 | −0.959324401 |
| charging technology | generic super charging stations | −0.691699732 | 0.691699732 | 0 | −0.646327877 |
| costs reduction | evs cost | 0.957140415 | −0.957140415 | 0.961179751 | 0.957801403 |
| costs reduction | the falling price of batteries | 0.957500995 | −0.957500995 | −0.929606356 | 0.932011183 |
| economic activity | economic activity | −1 | 1 | −0.985312975 | 0.990740486 |
| economic revene | business development | −1 | 1 | −0.957583063 | −0.959324401 |
| economic revene | current unit sales | −1 | 1 | −0.957583063 | −0.959324401 |
| economic revene | wealth | −1 | 1 | 0 | 0.121182442 |
| energy effeciency | energy efficiency | −0.952398323 | 0.952398323 | −1 | −0.959324401 |
| energy pollution | energy pollution | 0.388947408 | −0.388947408 | 0.796604556 | −0.510951584 |
| government regulation | carbon pricing | 0.156727117 | −0.156727117 | 0 | 0.121182442 |
| government regulation | cities conservation | 0.873254834 | −0.873254834 | −0.774093871 | −1 |
| government regulation | governments | 0.156727117 | −0.156727117 | 0 | −1 |
| government regulation | incentives | −0.691699732 | 0.691699732 | 0 | −1 |
| government regulation | public investment | −0.902626096 | 0.902626096 | 0.686233755 | −1 |
| government regulation | regulation | −0.691699732 | 0.691699732 | 0 | −1 |
| industry-university collaboration | scholarships | −0.952398323 | 0.952398323 | −0.957583063 | −0.959324401 |
| job creation | employment | −1 | 1 | −0.90171281 | 0.940099166 |
| motor technology | electric motor | 0.972982612 | −0.972982612 | 0.999909188 | 0.976648732 |
| other | a completely carbon neutral transportation option | −0.952398323 | 0.952398323 | −0.957583063 | −0.959324401 |
| industry-university collaboration | aboriginal training | −0.952398323 | 0.952398323 | −0.957583063 | −0.959324401 |
| usability | artificial intelligence | 0.156727117 | −0.156727117 | 0 | 0.121182442 |
| other | biomimicry | 0.156727117 | −0.156727117 | 0 | 0.121182442 |
| other | confidence | −0.952398323 | 0.952398323 | −0.957583063 | −0.959324401 |
| other | durability | 0.156727117 | −0.156727117 | 0 | 0.121182442 |
| other | environmentally conscious citizens | 0.156727117 | −0.156727117 | 0 | 0.121182442 |

**Table A1.** *Cont.*

| Categories | Concepts | Bad Economy | Good Economy | Battery Fail | Deregulation |
|---|---|---|---|---|---|
| usability | information technology | 0.156727117 | −0.156727117 | 0 | 0.121182442 |
| other | infrastructure | 0.96315824 | −0.96315824 | −0.774093871 | 0.995526376 |
| other | lack of hydrogen infrastructure | 0.924293982 | −0.924293982 | 0 | 0.915954432 |
| other | liability | 0.156727117 | −0.156727117 | 0 | 0.121182442 |
| other | oil and gas volatility | 0.156727117 | −0.156727117 | 0 | 0.121182442 |
| other | potential roadblocks | 0.957140415 | −0.957140415 | 0.937957076 | 0.957801403 |
| other | rare earth metals | 0.156727117 | −0.156727117 | 0 | 0.121182442 |
| other | remote communities | 0.156727117 | −0.156727117 | 0 | 0.121182442 |
| other | significant technology improvements | −0.957479374 | 0.957479374 | −0.060843278 | −0.957345752 |
| other | sustainability | 0.255551223 | −0.255551223 | 0.817909946 | 0.497931899 |
| other | the power and mileage limits | 0.924293982 | −0.924293982 | 0.961179751 | 0.915954432 |
| other | traffic congestion | 0.156727117 | −0.156727117 | 0 | 0.121182442 |
| other | transform mobility | −0.952398323 | 0.952398323 | −0.957583063 | −0.959324401 |
| public transportation | Self-driving vehicles | −0.74299687 | 0.74299687 | 0.542424672 | −0.816448312 |
| safety | economic and safety benefits | −1 | 1 | −0.957583063 | −0.959324401 |
| | **EV adoption** | **0.901968281** | **−0.901968281** | **0.798453798** | **0.889187005** |

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
