# Peer review of "Automatically Generating Scenarios from a Text Corpus: A Case Study on Electric Vehicles"

_sustainability, doi:10.3390/su14137938_

Round 1

Reviewer 1 Report

1. The study gaps need more clarification (the author should differentiate the proposed work from Feblowitz et al.( 2021).

2. On page 9, the author has compared the proposed model to the existing one (Kim et al.,2016). The advantages of defining questions over topic modelling need to be clarified.

3. Separate sections for study implications  are needed

4. Conclusion section should be expanded with a more detailed discussion on limitations and future scope.

5. More recent citations for 2021 and 2022  can be helpful.

Author Response

1. The study gaps need more clarification (the author should differentiate the proposed work from Feblowitz et al.( 2021).

>> We agree, this work was the closest to our approach. We have now detailed the differences at the beginning of the discussion. A key difference is that Feblowitz et al. used crowd-sourced questionnaires to get data on causal relationships, whereas that was part of our automation. 

2. On page 9, the author has compared the proposed model to the existing one (Kim et al.,2016). The advantages of defining questions over topic modelling need to be clarified.

>> Figure 3 shows that Kim et al. manually build their causal map, whereas we automatize this process. The main advantage is the automation, and the question of the paper was whether the resulting model is comparable with one produced manually. Defining questions over topic modeling in itself is not an advantage, but rather a consequence of a lack of automation. We hope that the clearer subdivisions of our discussion section will help readers in identifying our contributions, and we certainly remain available for any further clarification that may help.

3. Separate sections for study implications  are needed

>> The key sections for the paper are usually fixed (intro, background, methods, results, discussion, conclusion). However, we agree with the reviewer that it is beneficial for readers to quickly identify the key implications of this work. We thus flagged them via a new subdivision of the discussion. We also now emphasized that implications are not limited to electric vehicles.

4. Conclusion section should be expanded with a more detailed discussion on limitations and future scope.

>> Similarly to point #3 above, we used a new subdivision of the discussion to flag the limitations and future opportunities. We also expanded the descriptions for such opportunities. We note that the conclusion section itself could not be expanded, as it traditionally serves only to reiterate a few key takeaway points instead of providing information that would not be in a previous section.

5. More recent citations for 2021 and 2022  can be helpful.

>> We have added 15 references. As a result, we now have 19 references from 2021 and 12 references from 2022, that is, 31 references focusing on the last 18 months.

Reviewer 2 Report

1. Poor quality of the figures are presented 

2. I see that most of the paragraphs are cited with URL links and requesting the authors to cite properly as per journal guidelines. 

3. The numbering is missing for all the Equations 

4. Comparision is missing?

Author Response

1. Poor quality of the figures are presented 

>> We appreciate the reviewer’s keen attention. The figure files indeed had a low resolution. We have replaced them by high resolution files. Any compression artifacts would be caused by the publisher’s rendering of the article when creating a PDF.

2. I see that most of the paragraphs are cited with URL links and requesting the authors to cite properly as per journal guidelines. 

>> We apologize for missing this formatting requirement. We have now moved all URLs to references, including the access date.

3. The numbering is missing for all the Equations 

>> The two equations are now numbered, and cited by numbers in the text.

4. Comparision is missing?

>> The reviewer is right. There was a closely related work (Feblowitz et al) for which we did not provide a comparison. We have addressed this omission in the discussion, so that readers can now appreciate the distinction between this prior work and our approach. We note that our results section is entirely dedicated to a comparison, and is subdivided between structural comparison (i.e., content of the models) and scenario-based comparison (i.e., results from the models).

Reviewer 3 Report

The article was dedicated to a very interesting issue, which is  "Automatically Generating Scenarios From a Text Corpus: A Case Study on Electric Vehicles". The work discusses the problems of creating scenarios aimed at estimating the possible future. These scenarios are a key element in making decision-making processes, therefore, in the era of current automation and digitization, they constitute research issues that are much needed for examination.

Generally speaking, the article is interesting for the reader, it is also prepared properly in terms of the content. I only have a few comments that should be taken into account to make the article more valuable.

First of all, in the introduction part, please indicate whether, in addition to applying your scenarios to electric cars, it will also be possible to apply them to autonomous vehicles in the future? The article by Czech et al. for "Autonomous vehicles: basic issues." to which please refer to.

I have no objections to the methodological part.

In the discussion, it would be worth referring to the issue of education about electric vehicles, or rather the lack of it. It is worth referring to the article by Turoń et al. on "When, What and How to Teach about Electric Mobility? An Innovative Teaching Concept for All Stages of Education: Lessons from Poland". This may have a direct impact on the scenarios being developed. It is also an interesting research aspect that can be referred to in the future.

In summary, I am asking for its significant extension. Please, indicate the research limitations and future research plans of the authors. 

From the editing point of view, the article has a lot of errors that need to be corrected. First of all, it needs to be corrected to meet the mdpi requirements. References to the literature are incorrectly written - it should be numbered in square brackets [X]. Sequentially, please refer to the chapter numbering. References should not be alphabetical either, what is more it is incorectly formated, please pay attention to this.

In general, the article reads well and is interesting, so after making corrections I will be in favor of publishing it. Good luck!

Author Response

1. First of all, in the introduction part, please indicate whether, in addition to applying your scenarios to electric cars, it will also be possible to apply them to autonomous vehicles in the future? The article by Czech et al. for "Autonomous vehicles: basic issues." to which please refer to.

>> This is a good remark. Our tool could indeed be applied to other related problems in sustainability, or even other domains. Since this point is about broadening the use of the tool, we have added it in the discussion and cited the article by Czech et al.

2. In the discussion, it would be worth referring to the issue of education about electric vehicles, or rather the lack of it. It is worth referring to the article by Turoń et al. on "When, What and How to Teach about Electric Mobility? An Innovative Teaching Concept for All Stages of Education: Lessons from Poland". This may have a direct impact on the scenarios being developed. It is also an interesting research aspect that can be referred to in the future.

>> We agree that electric mobility education is an important concept pertaining to electric vehicles, so its omission from our study and the prior work had to be clarified. We have now pointed it out in the results, using the reference provided by the reviewer.

3. In summary, I am asking for its significant extension. Please, indicate the research limitations and future research plans of the authors. 

>> We had previously pointed out the limitations and future opportunities, but that section was buried in the discussion. We have now made it clearer through a dedicated subsection header, and we clarified our personal research plans. We also suggested one additional possibility for future work, in terms of using planners.

4. From the editing point of view, the article has a lot of errors that need to be corrected. First of all, it needs to be corrected to meet the mdpi requirements. References to the literature are incorrectly written - it should be numbered in square brackets [X]. Sequentially, please refer to the chapter numbering. References should not be alphabetical either, what is more it is incorectly formated, please pay attention to this.

>> We apologize for this omission. Although we have published in Sustainability previously, the formatting guidelines for references have clearly escaped our attention. We have reformatted all references in the [number] style, consecutively, and provided a DOI to help the publishing office. We look forward to working with the publication staff for any question related to references later on.

Round 2

Reviewer 1 Report

1. The revisions have been made according to the comment, but I have the following suggestion:

2. In the Discussion section, don't immediately start a subsection. (e.g. Findings and implications), instead, you can make a subsection Implication like you created a subsection "Limitations and opportunities for future studies."

Author Response

We appreciate the reviewer's keen attention to the structure of the manuscript. If we correctly understand the suggestion, it would replace a subsection header ("Findings and implications") by a subsection header ("Implication"). There are multiple implications, so we would still need to use the plural form. And implications stem 'from something'; here findings and followed by their implications. "Implications" would be an unusual and perhaps abrupt heading as it wouldn't be clear where the implications come 'from'. The situation may be different in other languages, where "Implications" would be a perfectly fine subsection header.

Reviewer 3 Report

Dear authors, the article has been significantly improved.  It looks great!  Thank you very much for the corrections.  The article is ready for publication in its current form.  All the best for you.

Author Response

We appreciate the kind words of the reviewer and we are pleased to hear that the article is "ready for publication in its current form". Thank you.